# Hybrid Silica-Coated PLGA Nanoparticles for Enhanced Enzyme-Based Therapeutics

**DOI:** 10.3390/pharmaceutics15010143

**Published:** 2022-12-31

**Authors:** Kyle T. Gustafson, Negin Mokhtari, Elise C. Manalo, Jose Montoya Mira, Austin Gower, Ya-San Yeh, Mukanth Vaidyanathan, Sadik C. Esener, Jared M. Fischer

**Affiliations:** 1Cancer Early Detection Advanced Research (CEDAR) Center, Knight Cancer Institute, Oregon Health & Science University, Portland, OR 97239, USA; 2Department of Biomedical Engineering, School of Medicine, Oregon Health & Science University, Portland, OR 97239, USA; 3Department of Electrical Engineering, Jacobs School of Engineering, University of California, San Diego, La Jolla, CA 92093, USA; 4Department of Bioengineering, Jacobs School of Engineering, University of California, San Diego, La Jolla, CA 92093, USA; 5Department of Nano Engineering, Jacobs School of Engineering, University of California, San Diego, La Jolla, CA 92093, USA; 6Department of Molecular and Medical Genetics, School of Medicine, Oregon Health & Science University, Portland, OR 97239, USA

**Keywords:** poly(lactic-co-glycolic acid), silica, nanoparticles, enzymes, amino acid depletion, cancer, drug delivery, double emulsion, biocompatible

## Abstract

Some cancer cells rely heavily on non-essential biomolecules for survival, growth, and proliferation. Enzyme based therapeutics can eliminate these biomolecules, thus specifically targeting neoplastic cells; however, enzyme therapeutics are susceptible to immune clearance, exhibit short half-lives, and require frequent administration. Encapsulation of therapeutic cargo within biocompatible and biodegradable poly(lactic-co-glycolic acid) nanoparticles (PLGA NPs) is a strategy for controlled release. Unfortunately, PLGA NPs exhibit burst release of cargo shortly after delivery or upon introduction to aqueous environments where they decompose via hydrolysis. Here, we show the generation of hybrid silica-coated PLGA (SiLGA) NPs as viable drug delivery vehicles exhibiting sub-200 nm diameters, a metastable Zeta potential, and high loading efficiency and content. Compared to uncoated PLGA NPs, SiLGA NPs offer greater retention of enzymatic activity and slow the burst release of cargo. Thus, SiLGA encapsulation of therapeutic enzymes, such as asparaginase, could reduce frequency of administration, increase half-life, and improve efficacy for patients with a range of diseases.

## 1. Introduction

Active enzymes offer attractive therapeutic strategies for many diseases [1,2,3,4,5], including cancer [6,7,8,9,10,11,12,13,14,15,16,17,18,19,20]. More specifically, amino acid depletion therapy (AADT) is a promising strategy that often employs active enzymes to target metabolic processes, which tumor cells depend upon for survival, growth, and proliferation [17]. Active enzymes for AADT (e.g., asparaginase, arginine deiminase, arginase, methioninase, cysteinase, etc.) deprive cancer cells of amino acids necessary for tumor growth [17]. Unfortunately, the immune response to foreign enzymes often prompts short elimination half-lives, which can limit clinical implementation of these therapies.

The linkage of polyethylene glycol (PEG) to an active drug or drug delivery vehicle (PEGylation) is one strategy developed to improve the half-life of therapeutic molecules including enzymes [21,22,23,24,25]. Table 1 includes several formulations of PEGylated enzymes (e.g., pegaspargase, calaspargase pegol-mknl, pegvaliase, pegargiminase, pegzilarginase, and PEG-KYNase) that are FDA-approved, undergoing clinical trials, or currently being researched. PEGylation shields enzymes from antibodies and proteases in vivo; however, the immune system can develop antibodies against PEG with repeated doses, reducing the long-term efficacy of PEGylated drugs [26,27,28,29,30]. In addition, a large percentage of the population has now been exposed to PEGylated liposomes during vaccination, thus increasing anti-PEG antibodies [31]. Patient sensitization to PEGylated formulations can complicate treatment regimens [32], highlighting an urgent clinical need for novel approaches to enzyme delivery. PEGylation can also reduce enzyme activity, further highlighting the need for alternative strategies for delivery [33].

Nanoparticles (NPs) offer high surface-area-to-volume ratios and loading efficiencies to improve the half-life of enzymes after delivery. PLGA NPs are of particular interest as they decompose into natural byproducts of human metabolism (lactic acid and glycolic acid) [34]. Several FDA-approved and commercially available therapies utilize PLGA particles for drug delivery [35,36,37,38,39,40], some of which are listed in Table 2. PLGA hydrolysis prompts the release of enzymatic cargo [41,42], which can begin hours after injection. While the rate of hydrolysis degradation can be tailored to an extent by modifying the molecular weight and monomer ratio of PLGA, dose dumping related to the characteristic burst release of enzymatic cargo from PLGA NPs remains a challenge for controlled release [34]. Surface modifications such as chitosan have been shown to help reduce the burst release of cargo from PLGA NPs [43,44,45]; however, burst release remains a major limitation for the controlled cargo delivery over extended periods.

Here we report a strategy to enhance retention of protein cargo by modifying the outer surface of PLGA NPs with porous silica. This hybrid drug delivery platform offers high enzyme loading efficiency, protection against proteolytic cleavage, and prolonged in-tissue stability. We outline the synthesis of enzyme-loaded core-shell PLGA NPs using a modified water-oil-water double emulsion technique [41] and the subsequent sol-gel condensation of an outer nanoporous silica layer [46,47,48,49]. The resultant SiLGA NPs permit diffusion of small molecule substrates into the NP core while protecting the enzymatic cargo from deactivation via proteinases and slowing burst release kinetics. Silica also offers unique surface modification strategies for targeted delivery. Both PLGA and silica are biodegradable and biocompatible, meaning that SiLGA NPs could have important clinical benefits as delivery vehicles for FDA-approved therapeutic enzymes like asparaginase.

## 2. Experimental Section

### 2.1. Preparation of Reagents

PLGA NPs were synthesized using a water-oil-water double emulsion technique. Fourteen milligrams of penicillinase (Penicillinase from Bacillus cereus, lyophilized powder, 1500–3000 units/mg protein using benzylpenicillin; P0389-25KU) from Sigma-Aldrich (St. Louis, MO, USA) were added to 450 µL of 1X PBS and vortex mixed (water phase 1). For in vivo experiments involving intramuscular injections, 2 mg of BSA-Cy7 (BS1-S7-1) from Nanocs, Inc. (New York, NY, USA) were added to 450 µL of 1X PBS and vortex mixed (water phase 1). To form the oil phase, 81 mg of PLGA (Resomer^®^ RG 504 H, Poly(d,l-lactide-co-glycolide), acid terminated, lactide:glycolide 50:50, Mw 38,000–54,000; 719900-1G) and 9 mg of DMT ((+)-Dimethyl l-tartrate, 99%; 163457-25G) were added to 3 mL of DCM (Dichloromethane, contains 40–150 ppm amylene as stabilizer, ACS reagent, ≥99.5%; D65100-500ML) and vortex mixed until solids visibly dissolved; all reagents were obtained from Sigma-Aldrich (St. Louis, MO, USA). The second water phase consisted of 5% by weight PVA (poly(vinyl alcohol) 87–90% hydrolyzed, average Mw 30–70 kDa; P8136-1KG) from Sigma-Aldrich (St. Louis, MO, USA) in milliQ water. Water was heated to 80 °C on a hotplate, and PVA was stirred into solution slowly. Once the PVA visibly dissolved at elevated temperature, the solution was vacuum filtered using the Stericup^®^ and Steritop^®^ Vacuum Driven Disposable Filtration System from MilliporeSigma (Burlington, MA, USA) and stored at 4 °C for at least one week prior to use.

### 2.2. Synthesis of PLGA NPs

PVA solution (water phase 2) was warmed to room temperature. Three hundred microliters of water phase 1 was added dropwise to the oil phase while vortex mixing on low. The resultant solution was vortex mixed on high for one minute to create the first emulsion (emulsion 1). The tube containing emulsion 1 was placed in ice water prior to probe sonication for two minutes at 30% amplitude (pulsed 2 s on and 1 s off) using the Q500 Ultrasonic Processor (#Q500) from QSonica (Newtown, CT, USA) at an output of 500 W at 20 kHz. A total energy of approximately 1400 J was transferred to emulsion 1, which consisted of aqueous nanodroplets (water phase 1; encapsulated penicillinase) suspended in oil (oil phase). Emulsion 1 was then immediately added dropwise to 18 mL of water phase 2 while stirring at 300 rpm to create the second emulsion (emulsion 2). The stir rate was increased to 600 rpm for 40 s. Emulsion 2 was transferred to a Falcon tube and vortex mixed for one minute on high. Emulsion 2 was then placed in ice water prior to probe sonication using the settings previously described. A total energy of approximately 2300 J was transferred to emulsion 2, which consisted of oil nanodroplets (oil phase) with an aqueous core (water phase 1; encapsulated penicillinase) that were suspended in an aqueous buffer (water phase 2). Emulsion 2 was capped with Parafilm™, vented with a syringe tip, and stirred at room temperature overnight at 200 rpm, allowing the DCM to evaporate out of the oil phase to yield solid PLGA NPs. PLGA NPs were placed in a vacuum desiccator for one hour. Approximately 16 mL of the initial 21.3 mL of emulsion 2 were recovered, suggesting that DCM evaporated out of solution.

### 2.3. Washing PLGA NPs

PLGA NPs were added to an empty ultracentrifuge tube (OPTI SEAL BELL TBE/56PK; 361625) from Beckman Coulter (Brea, CA, USA), which was then filled with milliQ water and mixed thoroughly. PLGA NPs were recovered via ultracentrifugation using the Optima Max-XP Table Top Ultracentrifuge from Beckman Coulter (Brea, CA, USA) at 32,000 rpm for thirty minutes at 4 °C. The supernatant was collected for downstream analysis, and the pellet was resuspended in 5 mL of milliQ water by sonication. This process was repeated three times. After the final wash, PLGA NPs were resuspended in 3 mL of milliQ water. PLGA NPs were then centrifuged at 1500 rpm for ten minutes at 4 °C. The supernatant was collected and sonicated. This step was repeated once. PLGA NPs were centrifuged a third time at 1500 rpm for thirty minutes at 4 °C. The supernatant was collected and sonicated.

### 2.4. Coating PLGA NPs with Silica

Half of the PLGA NPs (1.5 mL) were added to 30 mL of 0.1X PBS and vortex mixed thoroughly. The remainder of the PLGA NPs were stored at 4 °C. To form silicic acid, 74 µL of TMOS (tetramethyl orthosilicate, purum, ≥98% (GC); 87682-250ML) from Sigma-Aldrich (St. Louis, MO, USA) was added to 500 µL of HCl (0.1 mM). The silicic acid was vortex mixed thoroughly on high. While subsequently vortex mixing the suspension of PLGA NPs on low, 450 µL of silicic acid was added dropwise. Then, the suspension was vortex mixed thoroughly on high and shaken overnight at 4 °C to allow silicic acid to condense to silica on the surface of the PLGA NPs.

### 2.5. Washing SiLGA NPs

Approximately 13 mL of milliQ water was added to the SiLGA NPs. The resultant suspension was vortex mixed thoroughly and then centrifuged at 3900 rpm for fifteen minutes at 4 °C. The supernatant was collected for downstream analysis. The pellet was resuspended in one milliliter of milliQ water by pipetting and bath sonication for two minutes. Approximately 44 mL of milliQ water was added to the suspension of SiLGA NPs, which was then vortex mixed thoroughly. The SiLGA NPs were washed three more times using this technique. On the final wash, the pellet was resuspended in 1.5 mL of milliQ water and then stored at 4 °C.

### 2.6. Dynamic Light Scattering Measurements of PLGA and SiLGA NPs

PLGA and SiLGA NPs were sonicated to resuspend any particles that may have aggregated. One microliter of NPs was added to one milliliter of milliQ water. The resultant suspension was inverted ten times and loaded into a cuvette (folded capillary Zeta cell; DTS1070) from Malvern Panalytical (Malvern, UK). The cuvette was inserted into the Zetasizer Nano ZSP from Malvern Panalytical (Malvern, UK). Appropriate material properties were imported into the Malvern Zetasizer Software prior to measuring the hydrodynamic particle diameter, polydispersity index, and Zeta potential of the NPs. Three replicate measurements were obtained.

### 2.7. Nanoparticle Tracking Analysis of PLGA and SiLGA NPs

Samples were analyzed using the ZetaView QUATT Nanoparticle Tracking Video Microscope PMX-420 from Particle Metrix (Munich, Germany). The machine was calibrated with alignment beads (100 nm diameter) supplied by Particle Metrix. PLGA and SiLGA NPs were sonicated to resuspend aggregates that may have formed. NPs were diluted by a factor of at least 10^5^ into filtered (0.1 µm) milliQ water and 0.1X PBS to obtain particle counts within the liner range of the instrument. Samples were inverted ten times prior to being analyzed. Particle Metrix recommended the following settings, which were used consistently across samples: sizing and enumerating particles (70%); frame rate (30%).

### 2.8. Transmission Electron Microscopy of PLGA and SiLGA NPs

Transmission electron microscopy samples were prepared by spotting 10 µL of a 1:20 dilution of stock NPs in milliQ water on top of 200 mesh carbon film copper TEM grids (NC1513260) from Fisher Scientific (Waltham, MA, USA). Samples dried overnight. Micrographs were acquired using a Tecnai F20 TEM by FEI (Lausanne, Switzerland) at an accelerating voltage of 120 kV. Measurements of particle size were conducted manually using ImageJ.

### 2.9. Nitrocefin Assays Determine Penicillinase Activity

Penicillinase activity was measured as the linear rate of change in light absorption at 486 nm due to the catalysis of nitrocefin in vitro. The rationale for the assay was as follows: Active penicillinase shielded from PK degradation should catalyze nitrocefin, causing an increase in absorbance at 486 nm over time. When unprotected from PK degradation, penicillinase should not catalyze nitrocefin, and absorbance at 486 nm over time should therefore remain low. Five milligrams of nitrocefin (Nitrocein β-lactamase substrate; 484400-5MG) from EMD Millipore (Burlington, MA, USA) were thoroughly dissolved into 500 µL of DMSO. The dissolved nitrocefin was then added to 9.5 mL of 1X PBS to create a working solution, which was aliquoted, protected from light, and stored at −20 °C for up to two weeks. Aliquots of nitrocefin working solution were thawed at room temperature for at least thirty minutes prior to use in activity assays. Bare penicillinase, penicillinase-encapsulated PLGA NPs, and penicillinase-encapsulated SiLGA NPs were incubated with and without proteinase K (PK) prior to the addition of nitrocefin. PK was suspended in 1X PBS and added to penicillinase-containing samples at a final concentration of 0.1 mg/mL. Samples were incubated with PK for 4.5 h at 37 °C while shaking. Control samples were incubated with an equivalent volume of 1X PBS.

All samples were diluted into 1X PBS at a volume of 100 µL per well in an assay plate (96-Well Clear Bottom Plates with Lids, Tissue Culture Treated, White, Well Volume: 360 µL, Sterile; 3610) from Corning (Corning, NY, USA) and mixed thoroughly. Twenty-five microliters of nitrocefin working solution was added to each well at a total reaction volume of 125 µL per well. The assay plate was immediately withdrawn into a Tecan Spark^®^ 20M Te-cool™ (Männedorf, Switzerland) assay chamber set to 37 °C. Activity assays ran for at least one hour while shaking. Multiple (at least four) measurements per well of absorbance at 486 nm were averaged at each time point. Means of these average measurements were calculated across wells containing technical replicates and plotted against time. Penicillinase activity was calculated as the slope of the best-fit line through the linear region of those curves.

Nitrocefin absorbance shift assays were also used to estimate the loading efficiency and content of PLGA and SiLGA NPs. First, nitrocefin was added to a serial dilution set of penicillinase to generate a standard activity curve. Nitrocefin was then added to the wash buffers of PLGA and SiLGA NPs to measure unencapsulated penicillinase activity. Penicillinase concentrations of each wash buffer were estimated using the standard curve. Penicillinase concentrations were multiplied by the total volume of respective wash buffers and summed to yield the total amount of unencapsulated penicillinase.

### 2.10. Intramuscular Injection of PLGA and SiLGA NPs in Mice

Nod scid gamma (NSG) (005557) and albino C57Bl/6 (B6a) (000058) mice were purchased from the Jackson Laboratories. Mice were housed in high-efficiency particulate air (HEPA) cages in a specific-pathogen free (SPF) facility at OHSU. Mice were fed a diet of PicoLab Mouse Diet 20 (LabDiet, 5058) ad libitum beginning one week before imaging. Male and female mice were given a 50 µL intramuscular injection into each hind limb of Cy7-Bovine Serum Albumin (Nanocs) as free protein, PLGA encapsulated, or SiLGA encapsulated (*n* = 3 mice, or six total injection sites, for each treatment and mouse strain). One outlier injection site for the PLGA treatment in the NSG strain was removed from analysis. Mice were imaged for fluorescence (excitation wavelength = 740 nm, emission wavelength = 780 nm, exposure = 1 s) after correcting for background fluorescence using the IVIS Lumina XRMS Series III (PerkinElmer, Waltham, MA, USA).

## 3. Results

### 3.1. Structural Characterization of SiLGA NPs

We utilized a water-oil-water double emulsion technique for obtaining protein-loaded PLGA NPs [41] followed by silica coating to produce SiLGA NPs (Figure 1). Active enzymes suspended in water phase (“W1”) were interfaced with PLGA dissolved in oil (“O”). Emulsification at the W1/O interface created aqueous nanodroplets suspended in oil (“Emulsion 1”). Emulsion 1 was interfaced with a second water phase (“W2”). Emulsification at the O/W2 interface yielded an aqueous suspension of oil-phase NPs (“Emulsion 2”), which encapsulated enzymes in an aqueous core. The synthesis product of the W1/O/W2 double emulsion is shown in Appendix A. Evaporation of the oil phase formed an aqueous suspension of PLGA NPs with active enzymatic cargo. Subsequent sol-gel polycondensation of silicic acid formed silica layers around the PLGA NPs [46].

Dynamic light scattering (DLS), nanoparticle tracking analysis (NTA), and transmission electron microscopy (TEM) characterized the formation of complete and uniform silica layers around individual PLGA NPs. Under DLS, an increase in Cumulant particle diameter from 223 ± 2 nm (PDI: 0.099 ± 0.016) to 394 ± 26 nm (PDI: 0.252 ± 0.015) indicated deposition of silica layers (Figure 2a; data represent the average of technical triplicate measurements). Under NTA, NPs exhibited an average diameter of 162.7 ± 61.4 nm (*n* = 1687 particles) before silica coating and 187.7 ± 73.6 nm (*n* = 1501 particles) afterward (Figure 2b). DLS measured a shift in Zeta potential from −13.4 ± 0.1 mV prior to silica coating to −20.5 ± 1.4 mV afterward (Figure 2c; data represent the average of technical triplicate measurements). Zeta potential measurements using NTA were consistent with DLS, indicating a negative shift from −8.91 ± 0.26 mV to −17.10 ± 1.32 mV upon silica coating. Image analysis of TEMs revealed an average particle diameter of 92.1 ± 46.3 nm and an average coating thickness of 12.5 ± 4.3 nm (*n* = 58 particles). Representative electron micrographs of uncoated and coated PLGA nanoparticles are respectively shown in panels d and e of Figure 2. Silica at the surface of PLGA NPs can be observed as the change in contrast at particle edges [49]. These results showed that complete porous silica layers with uniform thickness are formed around PLGA NPs.

### 3.2. Enzyme Loading of SiLGA NPs

Enzyme loading efficiency is defined as the amount of enzyme encapsulated in NPs divided by the amount of enzyme added during synthesis, typically expressed as a weight percentage [50]. Enzyme loading content is defined as the amount of enzyme encapsulated in NPs divided by the starting mass of the NPs [50]. We measured the loading efficiency and content of PLGA and SiLGA NPs that encapsulated the enzyme, penicillinase. The majority (60%) of penicillinase lost during the synthesis was recovered in the wash buffer of precursor PLGA NPs; the remainder was recovered in the wash buffer of SiLGA NPs (Figure 3e). The enzyme loading efficiency of PLGA NPs was calculated to be approximately 94% and the enzyme loading content of the PLGA NPs was approximately 9.8%. The enzyme loading efficiency of SiLGA NPs was approximately 86% and the enzyme loading content was approximately 8.9%.

We compared the loading efficiency and content of our hybrid SiLGA NPs to reported values for chitosan-coated PLGA NPs, which demonstrably slowed the burst release of a variety of encapsulated cargo including SN38, ferulic acid, epirubicin, and paclitaxel [43,44,45,51]. SiLGA NPs achieved a similar loading efficiency (86% versus 87.1%) and content (8.9% versus 6.42%) to the optimal CS-PLGA NP formulations reported by Lu et al. [43]. SiLGA NPs exhibited higher loading efficiency (86% versus 74.17 ± 9.21%) and content (8.9% versus 4.2 ± 0.82%) than CS-PLGA NPs synthesized by Alibolandi et al. [51]. SiLGA NPs also showed higher loading efficiency than CS-PLGA NPs reported by de Lima et al. [45] (50 ± 4%) and Chen et al. [44] (72.0 ± 3.5%). These results show that the penicillinase-loaded SiLGA NPs had a loading efficiency that was similar to, if not higher than, those of reported drug-loaded hybrid CS-PLGA NP formulations intended to slow burst release kinetics.

### 3.3. SiLGA NPs Protect Enzymes from Proteolysis In Vitro

Proteolytic degradation of PEGylated enzymes has been suggested as a means to reduce therapeutic efficacy in cancer patients [52]. With systemic and local delivery being common modes of administration, enzyme-based therapies are susceptible to proteolysis from a variety of blood and tissue proteases. Viable encapsulation platforms should shield enzymatic cargo from proteolysis while retaining enzyme activity. We hypothesized that pores in the silica layer would be sufficiently small to prevent inward diffusion of proteases and outward diffusion of enzymatic cargo, yet large enough to permit inward diffusion of small-molecule substrates for catalysis.

To test the ability of SiLGA NPs to shield enzymatic cargo from proteolysis while retaining enzymatic activity, unencapsulated and encapsulated formulations of penicillinase were incubated with PK (large molecule; 28,900 g/mol) and nitrocefin (small molecule; 516 g/mol) to measure enzymatic activity of the intact NPs.

SiLGA NPs were notably more effective than PLGA NPs at protecting encapsulated penicillinase from proteolysis via PK (Figure 3a). Activity retention of the SiLGA formulation (46.7 ± 2.9%) after incubation with PK was approximately tenfold higher (*p* = 0.00002) than PLGA (4.33 ± 0.58%), which exhibited activity retention on par with unencapsulated penicillinase (10.2 ± 2.7%; “Bare”). Given that activity retention of the unencapsulated formulation was slightly greater than that of the PLGA formulation, a longer incubation period, a lower concentration of unencapsulated penicillinase, and/or a higher concentration of PK may have been necessary. Enzymatic activity levels were determined from representative curves that show shifts in optical absorbance as penicillinase in unencapsulated (Figure 3b), PLGA (Figure 3c), and SiLGA (Figure 3d) formulations catalyze nitrocefin before (black) and after (red) PK treatment. These results suggest that the silica coating protects loaded enzymes from large molecules, while still allowing for the passage of small molecules.

PLGA degrades in aqueous solutions as ester linkages between glycolic and lactic acid monomers hydrolyze [34,53]. Cargo release from PLGA particles generally undergoes three phases driven by hydrolysis degradation: an initial burst release of cargo, a sustained “lag” period with minimal release, and a final burst release [53,54]. Full hydrolysis of PLGA particles can span mere days to several months depending on factors including storage temperature, pH, and monomer ratios [34,53]. In this study, we used a PLGA formulation called Resomer^®^ RG 504 H with a 50:50 monomer ratio, which is known to exhibit one of the fastest hydrolysis rates amongst formulations of similar molecular weight [34,42]. This ratio suggests that our particles may degrade relatively quickly, perhaps closer to a matter of days rather than months. Importantly, elevated temperatures approaching or exceeding the glass transition temperature can also increase the rate of PLGA hydrolysis [53,55]. It is therefore likely that our in vitro and in vivo incubations at 37 °C expedited hydrolysis, considering that the manufacturer reported the glass transition temperature of Resomer^®^ RG 504 H to be less than 15 °C higher (46 °C to 50 °C).

Data from the literature suggest that noticeable hydrolysis degradation of our PLGA nanoparticle formulations in vitro and in vivo may occur over the course of one week post synthesis. The average molecular weight of a similar formulation of acid-terminated PLGA microspheres (50:50 monomer ratio; 21,000 g/mol) in an aqueous buffer at 37 °C decreased by approximately 50% over the course of one week due to hydrolysis degradation [56]. While Resomer^®^ RG 504 H exhibits a higher total molecular weight, and ostensibly a slower rate of hydrolysis, the cited authors noted that “…the molecular weight effect was considerably smaller than the end group effect…”. Data also suggest that amongst formulations with a monomer ratio of 50:50, acid-terminated PLGA degraded between two and four times faster than ester-terminated formulations [56]. Additionally, the authors stated that in vivo degradation occurred approximately twice as quickly as degradation in vitro, suggesting that a measurable degree of hydrolysis is feasible one day after injecting our formulation. It should also be noted that nanoparticles in our presented work exhibited a higher surface area-to-volume ratio than microparticles in the cited work. Consequently, a higher percentage of PLGA molecules in our formulation interfaced with the surrounding aqueous environment, possibly offsetting differences in hydrolysis rates due to molecular weight.

### 3.4. SiLGA NPs Slow the Burst Release of Enzymatic Cargo In Vivo

Finally, we tested whether silica can reduce the rates of PLGA hydrolysis and the subsequent burst release of cargo in vivo. For these experiments, we injected unencapsulated and encapsulated formulations of bovine serum albumin-conjugated to Cy7 (BSA-Cy7) intramuscularly into immunocompetent B6a (Figure 4a) and immunodeficient NSG mice. The BSA-Cy7 was meant to mimic the loading of an enzyme and the fluorescent signal allowed us to monitor the loss in real time with IVIS imaging. We chose intramuscular injection for our experiments because it is a common delivery mode for asparaginase in the treatment of acute lymphoblastic leukemia (ALL). PLGA and SiLGA NPs exhibited an initial burst release of BSA-Cy7 over the first two days, followed by a sustained lag period of minimal release through the fifth day and beyond in both immunocompetent (Figure 4b) and immunodeficient (Figure 4c) mice. Importantly, there was a significant difference (*p* < 0.05) between PLGA and SiLGA in terms of intramuscular levels of BSA-Cy7 after one day in both immunocompetent (Figure 4d) and immunodeficient (Figure 4e) mice, suggesting that silica slowed the initial burst release kinetics of protein cargo. After the second day, there was no difference between PLGA and SiLGA, suggesting that silica coating did not improve long term survival of encapsulated BSA-Cy7; however, both PLGA and SiLGA NPs successfully increased intramuscular retention of BSA-Cy7 relative to unencapsulated BSA-Cy7 for at least five days post-injection (*p* < 0.05). These results demonstrate that SiLGA NPs slowed burst release kinetics of enzymatic cargo while protecting the payload from immune clearance and proteolysis for an extended period.

## 4. Discussion

Here we demonstrate a new nanoparticle formulation for the encapsulation of active enzymes with a porous silica layer allowing small molecules to penetrate while protecting the enzymes from proteases and other large molecules. These nanoparticles can be produced in a consistent manner with small sizes, high enzyme loading, and improved biostability. Discrepancies in measurements of particle size across DLS, NTA, and TEM are to be expected. Reported values for particle diameter were lowest under TEM (92.1 ± 46.3 nm), followed by NTA (162.7 ± 61.4 nm) and DLS (223 ± 2 nm). Input samples for DLS and NTA were aqueous suspensions of nanoparticles feasibly swollen due to hydration, whereas nanoparticle suspensions were dried onto a substrate for TEM analysis. It follows that measurements from DLS and NTA characterized swollen nanoparticles with larger diameters than those observed under TEM due to differences in particle hydration. It is also feasible that samples for DLS, NTA, and TEM would be susceptible to nanoparticle aggregation during measurement. For DLS and NTA, the extent of nanoparticle aggregation can be difficult to quantify, and the impact of aggregation on measured values often goes uncorrected. In DLS, an aggregate may scatter disproportionately more light than the sum of its individual constituent particles, thereby inflating estimates of particle size. Metrics like polydispersity index (PDI) shed some light onto particle aggregation, though using PDI to differentiate particle aggregates from a population of individual particles with a wide size distribution remains difficult. While nanoparticle aggregation also occurs as suspensions dry onto TEM substrates, aggregation does not typically affect measurements from micrographs. This is due to the fact that boundaries between individual constituent particles within aggregates can be visualized under TEM. It is therefore feasible that DLS and NTA reflect larger particle diameters than TEM due in part to differences in sample hydration and susceptibility to aggregation during measurement.

Our enzyme loading contents of approximately 10% are common for inert drug carriers [50]. Loading content suggested that encapsulated enzymes could be delivered at approximately one-tenth the concentration of unencapsulated enzymes, assuming equal administered masses. While higher loading content would reduce the number of NPs necessary to deliver therapeutic enzymes and the likelihood of systemic toxicity, SiLGA offers potential advantages that could offset the reduction in deliverable enzyme concentration by prolonging enzyme activity.

The partial reduction in penicillinase activity after PK treatment of SiLGA NPs leads to several explanations. It is possible that pores in the silica layer were small enough to slow, but not completely prevent, the inward diffusion of PK or the outward diffusion of penicillinase to reduce, but not eliminate, enzyme proteolysis. Alternatively, SiLGA NPs may have released a portion of penicillinase cargo during storage that contributed to nitrocefin catalysis. In this scenario, PK would have proteolytically cleaved the unencapsulated penicillinase, explaining the reduction in activity retention after PK treatment. This explanation may also apply to activity retention of PLGA NPs, which are known to be susceptible to degradation via hydrolysis. If true, this explanation may suggest that the ability of the silica layer to shield enzymatic cargo may be at least partially related to improved NP stability in aqueous solution.

Overall, our data confirmed that SiLGA NPs successfully shielded penicillinase cargo from proteolysis without preventing the inward diffusion of small molecule substrates. The retention in penicillinase activity after PK treatment of SiLGA NPs is significant because it suggested the silica layer might have slowed PLGA degradation and the subsequent release of penicillinase cargo. Importantly, this interpretation of data is consistent with reported reductions in rates of cargo release from CS-PLGA NPs, which broadly suggested that an outer protective layer could potentially improve the half-life of PLGA-encapsulated enzymes by slowing the characteristic burst release of cargo. The combined features of proteolytic protection and controlled cargo release could be beneficial in the clinical administration of therapeutic enzymes.

Silica polycondensation reactions have been demonstrated to yield porous nanoparticle coatings [47,48,49]. One protocol relied on the diffusion of solvent molecules through the porous silica layer to dissolve the template particle, yielding hollow silica nanoparticles [49]. Another protocol utilized microporous silica to seal the nanoparticle surface, trapping relatively large (>2 nm) enzymatic cargo inside while permitting inward diffusion and subsequent catalysis of small molecule substrates [47]. Notably, we employed a similar chemical reaction in the presented work: mixing TMOS and HCl to form a silicic acid precursor that undergoes a polycondensation reaction, forming a porous silica layer at the nanoparticle surface [47,48,49].

Similarities between our results and published results suggest that: (1) silica layers form at the surface of PLGA nanoparticles; (2) silica layers are sufficiently porous to allow small molecule diffusion; and, (3) pores in the silica layer are sufficiently small to at least partially inhibit protein diffusion. The thickness of our silica coating (12.5 ± 4.3 nm; *n* = 58 particles) agrees with the range of 6 nm to 10 nm reported by Yang et al. [49]. Furthermore, SiLGA particles exhibited over 45% retention of enzymatic activity after treatment with the proteolytic enzyme, proteinase K, compared to less than 10% retention without coating. Ortac et al. also observed appreciable retention of activity after treating enzymes encapsulated in microporous silica-coated nanoparticles with proteolytic enzymes [47]. 

These results have two relevant implications. Firstly, small molecule substrates likely diffuse across the silica layer, suggesting the existence of pores. Secondly, a sufficient percentage of pores in the silica layer are likely small enough to at least partially protect enzymatic cargo from proteolytic cleavage. This characterization of pore size is also consistent with in vivo data, which suggests the silica layer reduces the rate of PLGA hydrolysis one day after intramuscular injection (Figure 4d,e). If nonporous, the silica layer may have entirely prevented PLGA hydrolysis as well as the outward diffusion of protein cargo shortly after injection. Instead, the notions that pores exist in the silica layer and that a subset of pores may be sufficiently large to permit protein diffusion are consistent with the interpretation that inward diffusion of water molecules prompt hydrolysis of PLGA, subsequently releasing BSA-Cy7 cargo. The potentiality that, on average, the pores are sufficiently small to limit protein diffusion is consistent with retention of enzymatic activity in vitro and levels of BSA-Cy7 one day after injection in vivo.

The controlled release of enzymes into circulation could be highly beneficial for patients with ALL. ALL blasts are auxotrophic for asparagine (Asn), meaning they do not express Asn synthetase and cannot synthesize Asn [57]. Asn is critical for cancer cell proliferation. As such, ALL blasts depend on environmental Asn for survival. A variety of asparaginase-based medications are prescribed to ALL patients, targeting environmental Asn. Asparaginase starves the tumor of a metabolic nutrient that is essential to their growth in a type of treatment called amino acid depletion therapy.

Elspar is the commercial name for *E. coli*-derived asparaginase, an FDA-approved enzyme used to treat ALL patients. Elspar requires a taxing administration of three times per week via intramuscular or intravenous injection. PEGylated versions of Elspar, like Oncaspar (pegaspargase) and Asparlas (calaspargase pegol), improve the elimination half-life of asparaginase and reduce the frequency administration. Unfortunately, many patients develop sensitivities to PEG and/or *E. coli*-derived asparaginase. Sensitized patients must follow desensitization regimens in order to continue asparaginase treatment [32]. Suggested desensitization regimens often involve asaparaginase erwinia chrysanthemi, the active enzyme in FDA-approved treatments like Erwinaze and Rylaze, which were initially developed for patients who develop allergic reactions or other sensitivities to *E. coli*-derived asparaginase.

SiLGA NPs could significantly enhance asparaginase-based treatments in the fight against ALL. Encapsulation of asparaginase within SiLGA NPs would ostensibly slow burst release after intramuscular injection, prolonging enzyme half-life and reducing the frequency of administration. Any of the FDA-approved asparaginases could ostensibly be loaded into SiLGA NPs using the water-oil-water double emulsion and sol-gel silica deposition described here. It is unlikely that PEGylation of asparaginase would prohibit encapsulation with SiLGA as PEG is soluble in water. A hypothetical formulation of pegaspargase or calaspargase pegol encapsulated within SiLGA NPs might benefit from both a slow burst release at the injection site and offer some protection against asparaginase proteolysis via proteases.

SiLGA NPs offer new strategies for asparaginase desensitization as well as a more manageable administration schedule for patients, potentially improving quality of life. Importantly, successful encapsulation of asparaginase in PLGA NPs has been demonstrated [58,59]. The benefits of coating PLGA particles with silica could extend to asparaginase as well as a wide variety of other therapeutic enzymes and particle formulations shown in Table 1 and Table 2. Future work will examine the impact of silica coating on the burst release kinetics of different molecular weights and monomer ratios of PLGA, the use of gatekeeper systems [47,60,61] to further control the release of enzymatic cargo, and the surface functionalization of silica for targeted delivery.

## 5. Conclusions

In this study, we introduce a drug delivery platform for the protection and controlled release of enzymes. Enzymes are encapsulated in PLGA NPs using a water-oil-water double emulsion technique. PLGA NPs act as scaffolds onto which uniform silica layers are deposited to form SiLGA NPs. Successful coating of silica was confirmed using TEM, DLS, and NTA. Loading efficiency and content were on par with previous PLGA hybrid NPs. SiLGA NPs offer encapsulated enzymes substantial protection against proteolytic cleavage in vitro compared to PLGA NPs. The addition of silica also slows the initial burst release of enzymatic cargo in vivo after intramuscular injection. Furthermore, encapsulated enzymes retain appreciable activity, catalyzing small molecule substrates that diffuse through the porous silica layer. The advantages of SiLGA NPs could easily extend to a variety of FDA-approved therapeutic enzymes that exhibit a short elimination half-life in vivo. Future work will examine the impact of polymer molecular weight, polymer monomer ratios, and gatekeeper systems on the controlled release of enzymatic cargo from SiLGA NPs.

## Figures and Tables

**Figure 1 pharmaceutics-15-00143-f001:**
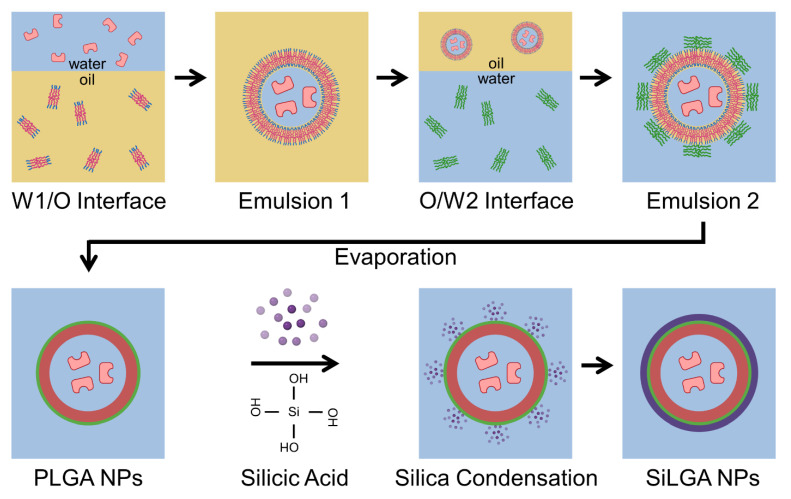
Synthesis of PLGA and SiLGA nanoparticles: Enzyme-encapsulated SiLGA NPs were synthesized using a water-oil-water double emulsion (PLGA NPs) and subsequent sol-gel polycondensation reaction of silicic acid to silica. Enzyme molecules (pink) were suspended in the first water phase (“W1”; blue). PLGA molecules (red and blue) were dissolved in an organic phase (“O”; yellow). The two phases were emulsified to form “Emulsion 1”, consisting of a suspension of aqueous droplets in oil. Emulsion 1 was then introduced to a second water phase (“W2”; blue) containing PVA molecules (green). A second emulsification produced “Emulsion 2”, which consisted of an aqueous suspension of oil droplets with an aqueous core. Subsequent evaporation of the oil phase solvent yielded an aqueous suspension of PLGA NPs (red) with an enzymatic core (pink) that was stabilized by PVA (green). Subsequent introduction of silicic acid (purple dots) initiated a polycondensation reaction, forming a porous silica layer (purple) at the surfaces of the PLGA NPs to form SiLGA NPs. A portion of this figure was created using Biorender.com (BioRender 2021), accessed on 18 October 2021.

**Figure 2 pharmaceutics-15-00143-f002:**
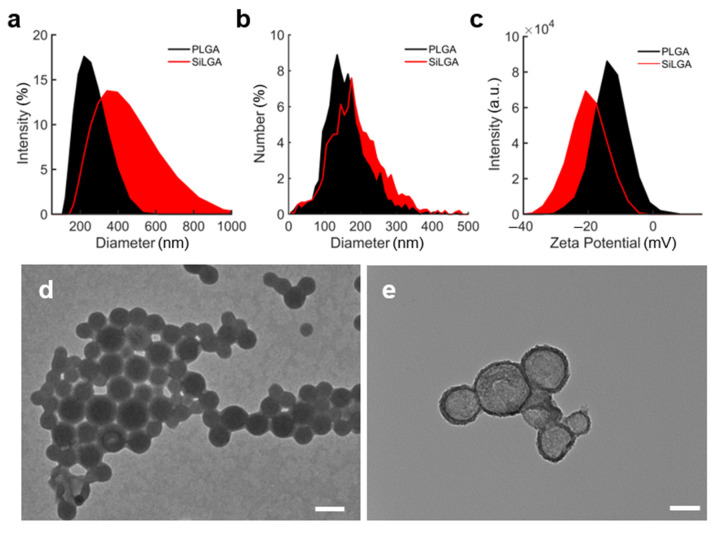
DLS, NTA, and TEM show silica coating: Distributions of (**a**) Cumulant hydrodynamic diameters, (**b**) hydrated diameters, and (**c**) Zeta potential of PLGA (black) and SiLGA (red) NPs. Transmission electron micrographs of (**d**) PLGA NPs (scale bar is 200 nm) and (**e**) SiLGA NPs (scale bar is 100 nm).

**Figure 3 pharmaceutics-15-00143-f003:**
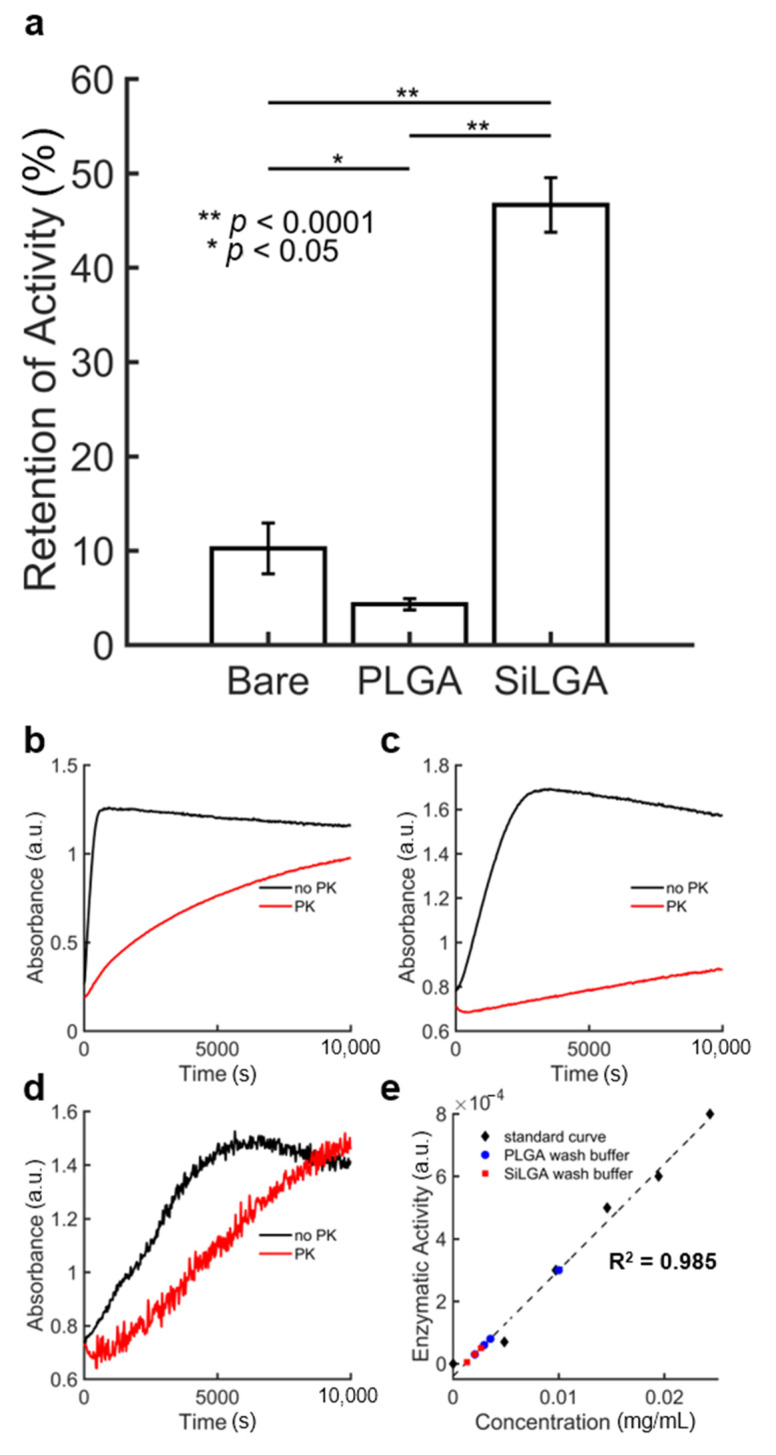
Penicillinase loading and retention of activity after treatment with PK: (**a**) The percent retention of penicillinase activity for unencapsulated (“Bare”) and encapsulated (“PLGA”; “SiLGA”) formulations after treatment with the proteolytic enzyme, PK. Error bars represent one standard deviation of the mean across triplicate measurements. Representative curves shows the shift in optical absorbance when nitrocefin substrate is added to penicillinase in (**b**) unencapsulated, (**c**) PLGA, and (**d**) SiLGA formulations both before (black) and after (red) PK treatment. Each contour represents the average of technical duplicate measurements. (**e**) Standard curve for enzymatic activity as a function of penicillinase concentration (black diamonds; R^2^ = 0.985). Measured penicillinase activity of wash buffers for PLGA (blue circles) and SiLGA (red squares) NPs were plotted to estimate respective penicillinase concentrations for loading efficiency and content calculations.

**Figure 4 pharmaceutics-15-00143-f004:**
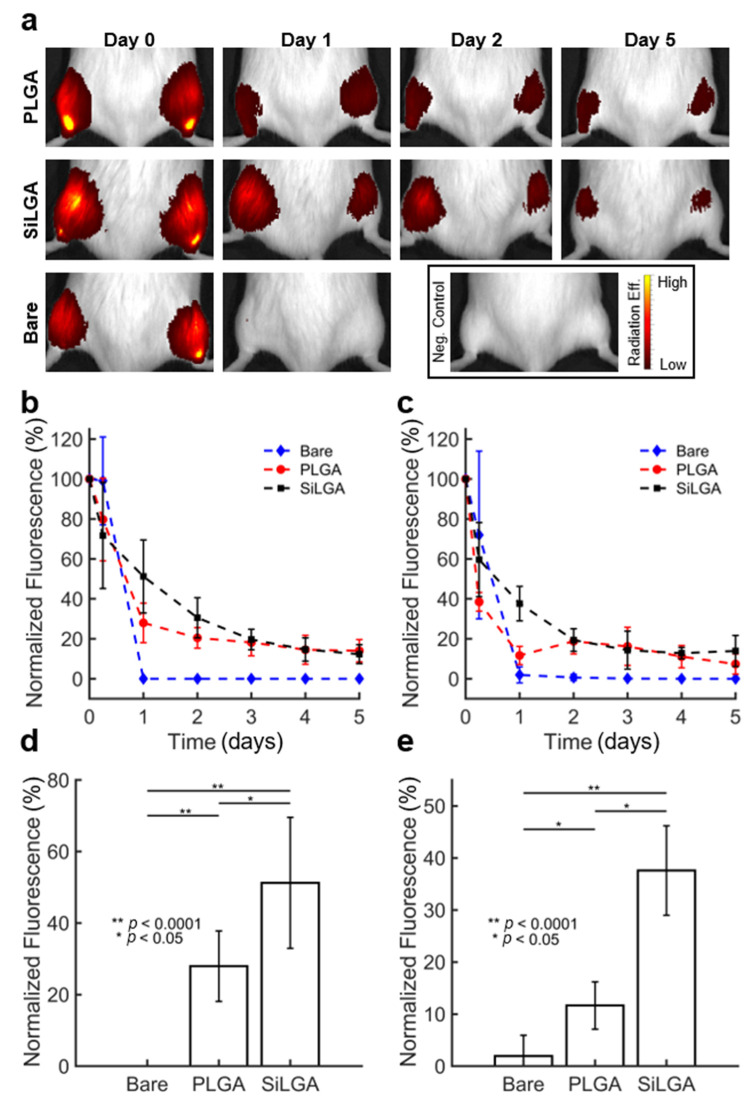
Silica slows burst release kinetics of BSA after intramuscular injection in mice: (**a**) Radiation efficiency of BSA-Cy7 PLGA NPs (**top**), BSA-Cy7 SiLGA NPs (**middle**), and unencapsulated BSA-Cy7 (**bottom**) in the hind legs of immunocompetent mice (B6a) as imaged over a five-day span; saline was injected into the negative control shown at Day 0. Normalized fluorescence levels of intramuscular BSA-Cy7 over a five-day span in (**b**) B6a and (**c**) NSG mice. Error bars represent one standard deviation above and below the mean normalized fluorescence level, calculated across six injection sites. (**d**) Normalized fluorescence levels of unencapsulated (“Bare”) PLGA, and SiLGA formulations of BSA-Cy7 after one day in (**d**) B6a and (**e**) NSG mice.

**Table 1 pharmaceutics-15-00143-t001:** Active enzymes used to treat diseases.

Active Enzyme	Enzyme Formulation(s)	FDA Status	Disease(s)	Reference(s)
asparaginase	colaspase; crisantaspase; pegaspargase; calaspargase pegol-mknl	approved	acute lymphoblastic leukemia; lymphoblastic lymphoma	[5,6,7,8,9,10,17,21,22,24]
phenylalanine ammonia lyase	pegvaliase	approved	phenylketonuria	[2,3,4,5]
arginine deiminase	pegargiminase; ADI-PEG 20	phase 2	metastatic melanoma; hepatocellular carcinoma; mesothelioma	[5,11,12,13,14,17,18]
arginase	pegzilarginase	phase 3; phase 1	Arginase 1 deficiency; melanoma, leukemia, lymphoma	[1,5,17]
methioninase		phase 1	lung carcinoma, breast cancer, renal cancer, lymphoma	[5,19,20,23]
kynureninase	PEG-KYNase	preclinical research	melanoma, breast carcinoma, colon carcinoma	[15]
cysteinase		preclinical research	prostate cancer, breast cancer	[16]

**Table 2 pharmaceutics-15-00143-t002:** Commercial formulations using PLGA particles for drug delivery.

Commercial Name(s)	Active Ingredient	Disease(s)	Reference(s)
Decapeptyl^®^; Decapeptyl™ SR; Trelstar™ Depot	triptorelin	prostate cancer, endometriosis	[36,37,38,39]
Bydureon^®^; Bydureon Bcise^®^	exenatide	type 2 diabetes	[35,39,40]
Sandostatin LAR^®^; Sandostatin LAR^®^ Depot	octreotide	acromegaly	[36,38,39]
Suprecur^®^ MP	buserelin	prostate cancer, endometriosis	[36,38]
Somatuline^®^ LA	lanreotide	acromegaly	[36,38,39]
Lupron Depot^®^	leuprolide	prostate cancer	[36,38,39]
Nutropin Depot^®^	somatropin	pediatric growth hormone deficiency	[36,38]

## Data Availability

The data that support the findings of this study are available from the corresponding author upon reasonable request.

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
