# Peer review of "Hybrid Silica-Coated PLGA Nanoparticles for Enhanced Enzyme-Based Therapeutics"

_pharmaceutics, 2022, doi:10.3390/pharmaceutics15010143_

Round 1

Reviewer 1 Report

Silica-coated PLGA nanoparticles were prepared from double emulsion evaporation and further modification with a silica sol. These particles were characterized sufficiently. High loading efficiency and content of an enzyme have been shown. A slower release of the cargo was investigated in vivo via fluorescence imaging. The manuscript is of good quality.

My main concern is about the slow burst release from the coating PLGA nanoparticles. It was mentioned that the silica pores were small enough to keep large biomolecules from going through, but were large enough to allow inward diffusion of small molecules. Silica is generally quite stable under physiological conditions. How would that allow BSA-Cy6 diffused out of the nanoparticles?

Also, the release of enzymes (as mentioned in the introduction) and other large biomolecules from PLGA is a result of PLGA being hydrolysed. I suggest to provide the hydrolysis data/conditions for RG504, either from your own research or from literature.

After PVA was visibly dissolved, what was removed during the filtration process? Was there any advantage to store the PVA solution for at least 1 week before use?

Figure 1: need to describe what those structures/symbols are in the scheme.

It would be good to have an image to show the W/O/W emulsion formed.

Encapsulation of BSA-Cy7 was not mentioned in the Experimental section.

Reviewer 2 Report

The authors present an enzymatic encapsulation system for drug delivery purposes. With biological drugs being in an explosive rise on the market, novel drug delivery systems that provide proper protection against the fragile biomolecules are urgently needed. Coating with a ceramic silica layer should provide such proper protection, which is also the rationale of the current study.

One crucial issue that remains unclear throughout the study is based on what the authors claim that the formed silica coating would be porous? There is no evidence presented that would indicate that that would be the case, and is is also unclear from the synthesis protocol why this would yield a porous layer. This evidence should be provided before publication since the authors make such a strong point around this claim. (Without this it also renders the future prospects of "gatekeeping" mechanisms void.)

Was image analysis performed on the TEM images, given that sizes are being derived from these?

Technical comments:

- It seems uncustomary to refer to Supporting Information in the Introduction (in the first sentence, nonetheless) and it disrupts the reading flow; the tables are not that large that they could not be included into the main text. References should be included into the tables though - now there is not one single reference in them.

- Using the required template would also improve the reading flow.

- Is it not customary for US-based universities to include country in the affiliations?
